# Impacts of Risk Perception and Environmental Regulation on Farmers' Sustainable Behaviors of Agricultural Green Production in China

**Mingyue Li, Yu Liu, Yuhe Huang, Lianbei Wu and Kai Chen ***

School of Economics and Management, Beijing Forestry University, Beijing 100083, China; limingyue_2019@bjfu.edu.cn (M.L.); liuyu0228@bjfu.edu.cn (Y.L.); huangyuhe@bjfu.edu.cn (Y.H.); wulianbei@bjfu.edu.cn (L.W.)
* Correspondence: chenkai3@bjfu.edu.cn

**Abstract:** In China, the excessive application and improper disposal of chemical inputs have posed a great threat to the agricultural ecological environment and human health. The key to solve this problem is to promote the sustainable behaviors of farmers' agricultural green production (AGP). Based on the micro-survey data of 652 farmers, this study adopts the binary probit model to investigate the impacts of risk perception and environmental regulation on the sustainable behaviors of farmers' AGP. Results show that both risk perception and environmental regulation have significant effects on farmers' willingness to engage in sustainable behaviors. Moreover, environmental regulation can positively adjust risk perception to improve farmers' willingness to engage in sustainable behaviors. In terms of the two-dimensional variables, economic risks create the greatest negative impacts, and their marginal effect is 7.3%, while voluntary regulation creates the strongest positive impacts, and its marginal effect is 14.1%. However, both constrained and voluntary regulation have an enhanced moderating effect, where the effects of voluntary regulation are more remarkable. This is mainly because the environmental regulation policy signed by the government and farmers through the letter of commitment can inspire farmers to continue to implement green agricultural production from the deep heart. Therefore, government policies should constantly reduce farmers' risk perception in terms of economic input, and adopt restrictive behaviors measures, such as regulatory punishment and voluntary contract, to promote their sustainable behaviors of AGP to the maximum extent.

**Keywords:** agricultural green production; farmers' sustainable behaviors; economic risks; voluntary regulation; binary probit model

## 1. Introduction

As the collective wisdom of human's long-term agricultural production practice grows, the massive application of chemical input, such as chemical fertilizers and pesticides, has continuously accelerated the process of world agricultural production [1]. However, such extensive development has caused serious ecological environment problems, such as soil structure destruction, underground water pollution and biodiversity decline. Moreover, it poses a great threat to the quality improvement of agricultural products and the maintenance of human health [2–4]. Therefore, people are increasingly becoming aware of the importance and urgency of promoting the transformation and upgrading of agricultural production to achieve green development. Agricultural green production (AGP) is a comprehensive category based on the concept of green development. In the actual production process, it involves soil and fertilizer management, pest and disease management technology, agricultural product storage management and sales, etc. [5]. Many studies have pointed out that AGP plays a positive role in solving the deteriorating ecological environment in agricultural production [6,7]. Chinese authorities have issued a series of documents, including 'Technical Guidelines on Agricultural Green Development

(2018–2030)' and 'Opinions on Comprehensively Promoting Rural Revitalization and Accelerating Agricultural and Rural Modernization'. They aim to replace the traditional extensive production mode with the green production mode characterized by energy saving, environmental protection and pollution reduction in order to realize the green development of agriculture. Nevertheless, the government still fails to effectively curb the deterioration of environmental pollution caused by excessive reliance on chemicals despite the efforts at all levels to promote AGP [8,9].

In fact, the key to solving the prominent problem of environmental pollution and truly realizing the green transformation and high-quality development of agricultural production lies in the application of farmers' sustainable behaviors of AGP [10]. Previous research mainly focused on whether farmers respond to the initial behaviors of AGP. Moreover, most of them focused on discussing the effects of basic characteristics of individuals and families (e.g., age, education level, household labor force and cultivated land scale) [11], the form of production organization (e.g., whether to join farmers' cooperatives) [12], social capital (e.g., relationship network, social trust, reciprocal cooperation, information attribute) [13,14], market environment (e.g., material input, market price, sales model) [14,15] and other factors on the initial behavior response. These results are very helpful for us to understand farmers' AGP behaviors, but there is still a lack of exploration on whether farmers will continue their AGP behaviors after the first implementation. As the extension of the initial behaviors continues over time, the sustainable behaviors of farmers' AGP are also worthy of attention. Only through the sustainable implementation of AGP behaviors can farmers have a positive and far-reaching impact on the traditional extensive agricultural production mode at the cost of wasting resources and polluting the environment [16,17].

In a few studies on the sustainable behaviors of farmers' AGP, some scholars explored the relationship and effects between different dimensions of expected confirmation and the sustainable behaviors of farmers' soil testing and formula fertilization technology by constructing an extended sustainable adoption model [17]. In addition, the study applied the theory of planned behaviors to analyze the impact of attitude, subjective norms and perceived behavioral control on farmers' sustainable agricultural production technology [10]. However, these literatures are all based on psychological factors, especially subjective perception variables, which makes it difficult to investigate the sustainable behaviors of farmers in AGP (which cannot fully reflect the complexity of farmers' sustainable behaviors). This is because farmers' behavioral decision-making is usually the result of the combined action of internal and external factors [18,19]. Existing studies also showed that the direct factor influencing farmers' environmental behavior is the objective situation faced by the subject, which is rooted in the inner constraint [20]. Therefore, in order to make up for the defects of existing literatures, this study will consider the analysis and demonstration from the two aspects of internal perception constraints and external environmental factors, which will help further explore the sustainable behaviors of farmers' AGP.

The sustainable behaviors may be affected by many factors. With the implementation of AGP action, the influence of internal perception level and external environmental factors on farmers' sustainable behaviors of AGP grows more and more obvious. As two important aspects of farmers' internal perception and external environmental factors, risk perception and environmental regulation provide a new perspective for the study of farmers' sustainable behaviors of AGP. On the one hand, risk perception can provide effective information and reduce uncertainty, thus prompt the subject to make reasonable decisions [21]. In agricultural production, the farmers improper behaviors of applying chemical inputs harms the soil eco-environment, the quality and safety of agricultural products and the health of human beings and livestock [22]. The risk perception internalized by farmers can enable them to fully predict the probability of their own behavioral risks, and then make a clear judgment on the disadvantage consequences of their behaviors [23,24]. On the other hand, green production can internalize the external diseconomy of agricultural environmental pollution. However, for farmers, engaging in AGP is an investment behavior [5,8]. The government needs to take institutional measures to ensure the capital

input required by farmers to improve the initiative and enthusiasm of farmers to continue their AGP. As a typical institutional means, environmental regulation can internalize the external diseconomies caused by non-green production behaviors, such as the abuse of chemical fertilizers and pesticides and the littering of waste in farmers' agricultural production [25]. Obviously, environmental regulation plays an important role in promoting farmers' sustainable behaviors of AGP.

To sum up, risk perception provides possible risk judgment and evaluation for the sustainable behaviors of farmers' AGP. Environmental regulation provides incentives or constraints for the sustainable behaviors and can adjust farmers' risk perception level under certain circumstances. Therefore, there are two main two research questions. First, what are the impacts of risk perception and environmental regulation on the sustainable behaviors of farmers' AGP? Second, what are the roles of environmental regulation in the relationship between risk perception and the sustainable behaviors of farmers' AGP? The two research questions were answered by identifying the following specific objectives: (1) unify risk perception and environmental regulation into this study and reconstruct the dimensional space of risk perception and environmental regulation; (2) investigate the direct impacts of risk perception and environmental regulation and their dimensions on the sustainable behaviors of farmers' AGP; (3) explore the moderating effects of environmental regulation on risk perception and the sustainable behaviors of farmers' AGP. The main contribution of this study is to determine the sustainable behaviors of farmers' AGP, and to investigate the key factors affecting the sustainable behaviors of farmers' AGP from the perspectives of internal perception and external environment. The research findings will provide a reference for decision makers to formulate and improve policy interventions to reduce eco-environmental degradation.

The rest of this study is arranged as follows: Section 2 introduces materials and methods, including survey samples and data collection, variable selection and measurement, economic model construction, etc. The results are presented in Section 3. Section 4 provides an in-depth discussion. Conclusions and related policy recommendations are presented in Section 5.

## 2. Materials and Methods

### 2.1. Study Area

The research area is located in Henan Province, China. This province is one of the major grain-producing provinces in the plain region of China, with a grain sown area of 14,741,610 hectares, accounting for 12.625% of the national grain sown area. The province can plant grains for two seasons a year, among which winter wheat is one of the major agricultural products. According to the Statistical Yearbook of Henan Province—2021, the total agricultural output value of Henan province in 2020 was 624.484 billion yuan, higher than that of other Chinese provinces. In addition, the total wheat output in the province was 37.531 million tons in 2020, accounting for 27.955% of the national wheat output and ranking first in China among provinces. However, the province is also facing great resource and environmental pressure, among which the excessive application of chemical inputs such as fertilizers and pesticides is an important source of agricultural non-point source pollution [5]. According to China's Rural Statistical Yearbook—2021, Henan province used 6.48 million tons of agricultural fertilizer in 2020, more than any other province in China, and the province used 102,400 tons of pesticides, second only to Shandong province (which ranked first in pesticide use (114,113 tons)). As one of the major grain-producing provinces, Henan plays a pivotal role in ensuring national food security. Nonetheless, the problems of agricultural eco-environment in this province are still outstanding.

### 2.2. Data Collection

The data used in this study were derived from a field survey of grain growers in the major grain-producing counties in China from September to November 2021. In order to ensure the rationality of this specific research points, this study selected three super

grain-producing counties from the grain-producing counties published by the financial department of Henan province in November 2020 (Department of Finance of Henan Provincial of China, 2020), namely, Hua County in Anyang City, Xiayi County in Shangqiu City and Taikang County in Zhoukou City (Figure 1). Due to the large population and good grain production conditions, the 3 sample counties receive special funds from the central finance of China every year. Meanwhile, drawing on research methods of Sharifzadeh et al. (2019) [26], Jiang et al. (2021) [14] and Mao et al. (2021) [27], this study adopts multi-stage sampling to collect sample data. Firstly, according to geographical location, two townships (towns) were selected from each sample county. Then, four administrative villages were selected from each sample township (town) according to the officially registered villages; Finally, 25–30 farmers were selected from each sample village. The selected sample farmers are all major family members with local household registration and engaged in agricultural production, which is the prerequisite for the investigation. Based on this information, 720 households in 24 administrative villages and 6 towns were selected as the research objects.

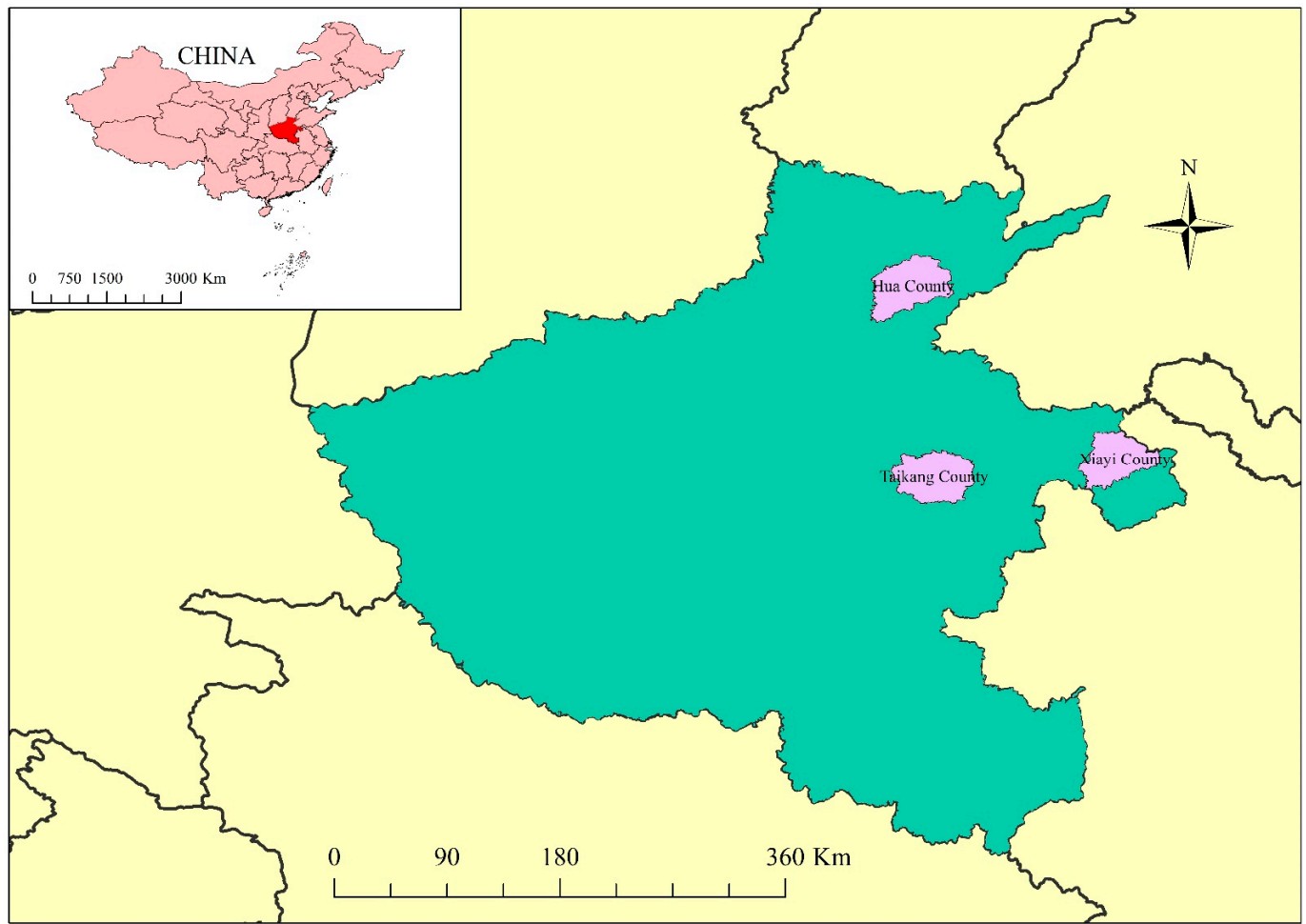

**Figure 1.** Location of the sample areas in this study.

The formal questionnaires were used to cover the basic characteristics of individuals and families, household production and operating conditions, AGP situation, farmers' risk perception and government environmental regulations. Before the formal questionnaire was confirmed, the research group specially invited two professors in related fields to discuss the questionnaire. Furthermore, 120 grain growers in Dancheng County outside the sample area were pre-surveyed. After careful and meticulous modification and improvement, the final formal questionnaire was determined, and its validity and clarity have been

significantly improved. In addition, considering the overall cognitive level of sample farmers, this study adopts the way of "face-to-face question and answer" between investigators and sample farmers to fill in the questionnaire. Before the formal investigation began, the research group also conducted unified training for investigators. Meanwhile, some local college students who could understand the local dialect were recruited to help fill out the questionnaire in order to avoid the errors caused by language differences as much as possible. After each questionnaire was filled in, packaged milk prepared in advance was delivered to each sample farmer in time as a token of gratitude. After removing invalid questionnaires such as missing key information, 652 points of valid questionnaires were collected in this study, with an effective rate of 90.6%. Among them, 213 were in Hua County, 245 in Taikang County and 194 were in Xiayi County.

*2.3. Variables and Measurement*

The sustainable behaviors of farmers' AGP were taken as the explained variable in this study, which are the continuous implementation of AGP behaviors in the time dimension, and specifically refers to the willingness of farmers to engage in green production from the initial to a long period of time in the future (from the initial application to the point of investigation). It is a kind of behavior after the response to green production, not only including the willingness but also the unwillingness to continue the implement. According to the actual situation of farmers and their families' agricultural production in the research region, and referring to the existing relevant research literatures [24,28], this study, on the basis of clarifying the above definition, sets the following questions in the questionnaire to measure farmers' sustainable behaviors: "Is your family willing to continue to implement AGP in the next period of time?". If the farmer answers "yes", the value is assigned to 1; Otherwise, the value is 0.

The core explanatory variables are farmers' risk perception and environmental regulation. Trujillo-Barrera et al. (2016) [29] showed that farmers' enhanced risk perception will not only directly reduce the opportunity of adopting sustainable practices but also weaken the role of expected economic returns brought by them. Because farmers must take into account the costs paid by themselves and their families (e.g., money and time) and the influence of external environment (e.g., ambient pressure) in their behavioral decision-making process [5]. Some studies also confirmed that farmers' environmental risk perception has a significantly direct impact on farmers' pesticide application behaviors using the indicators including the soil pollution, air pollution, water pollution and others [8,30]. Therefore, based on the research results of Pan et al. (2020) [31], Zhou et al. (2020) [32] and Mao et al. (2021) [27], risk perception in this study is defined as the judgment and evaluation of farmers' perceived economic, time, environment, pressure, etc., during their sustainable AGP. On this basis, this study combines the 5-level Likert Scale and adopts four dimensions of economic risks, time risks, environmental risks and pressure risks and assigns them values between 1~5 to measure the risk perception of farmers' sustainable behaviors.

While most of the existing literatures on environmental regulation investigates farmers' behaviors from the aspects of constrained regulation, incentive regulation and guiding regulation. Among them, constrained regulation mainly refers to mandatory intervention methods, such as supervision and punishment, to improve agricultural environmental quality and the utilization efficiency of input resources by strengthening the role of government [33,34]. Incentive regulation mainly refers to the market-oriented economic incentives, such as economic incentives and environmental subsidies, to promote the control and optimization of environmental pollution [35,36]. Guiding regulation mainly refers to the way of information distribution mechanism, such as publicity and promotion, education and training, to guide production subjects to participate in rural environmental governance (Zhang et al., 2018 [37]; Gao et al., 2020 [6]). There are also literatures concerning the influence of voluntary regulations on the subjects' active implementation of environmentally friendly behaviors, such as household waste recycling or harmless treat-

ment of livestock and poultry feces [38,39]. Hence, this study, drawing on the research results of Zhao et al. (2018) [40], Varela-Candamio et al. (2018) [36], Si et al. (2020) [39] and Du et al. (2021) [11], defines environmental regulation as a series of measures and policies that are formulated and implemented by government departments to improve the agricultural green development in response to negative external behaviors such as ecological destruction, environmental pollution and resource waste of farmers in agricultural production. Besides, four dimensions of constrained regulation, incentive regulation, guiding regulation and voluntary regulation are selected, and environmental regulation is measured from the perspective of the impacts of government policies and measures on farmers' behaviors. The 5-level Likert Scale is also adopted to assign and measure the four dimensions of environmental regulation.

Moreover, according to the research of Teklewold et al. (2013) [41], Zhang et al. (2018) [37] and Xie and Huang (2021) [42], other factors affecting the sustainable behaviors of farmers' AGP are selected as control variables, including gender, age, education level, arable area, share of non-farm income, household labor force and effectiveness evaluation. These factors can not only affect farmers' sustainable behaviors, but also distinguish the farmers willing to continue the implement from those who are not. It should also be mentioned that farmers' subjective understanding and feeling of AGP behaviors, as well as their judgment and recognition of the positive effects, are the premise for the continuous implementation of AGP behaviors [17]. Therefore, this study scores the farmers' responses to the 'evaluation of the overall effect after the first implementation of AGP behaviors', and assigns the effect evaluation variable as 1, 2, 3, 4 and 5, where 1 represents very bad and 5 represents very good.

The definitions and assignments of the explained and explanatory variables in this study are shown in Table 1.

**Table 1.** Measurement items of variables.

| Variables | Number | Definition and Assignment |
|---|---|---|
| Willingness to continue the implementation | Cin | Whether you are willing to continue to implement agricultural green production behaviors? If yes = 1, if no = 0 |
| Risk perception | Rpe | It was obtained by weighted average of Rpe1–Rpe4 |
| Economic risks | Rpe1 | Do you worry that more money will be spent to implement agricultural green production behaviors? Strongly disagree = 1, disagree = 2, neutral = 3, somewhat agree = 4, strongly agree = 5 |
| Time risks | Rpe2 | Do you worry that it will cost more time and energy to implement agricultural green production behaviors? Strongly disagree = 1, disagree = 2, neutral = 3, somewhat agree = 4, strongly agree = 5 |
| Environmental risks | Rpe3 | Do you worry that the eco-environment damage will be aggravated if the agricultural green production behaviors is no longer implemented? Strongly disagree = 1, disagree = 2, neutral = 3, somewhat agree = 4, strongly agree = 5 |
| Stress risks | Rpe4 | Do you worry that your neighbors will talk more about implement agricultural green production behaviors? Strongly disagree = 1, disagree = 2, neutral = 3, somewhat agree = 4, strongly agree = 5 |
| Environmental regulation | Ere | It was obtained by weighted average of Ere1–Ere4 |
| Constrained regulation | Ere1 | How much is farmers' agricultural green production behaviors affected by government regulation and punishment policies? No effect = 1, small effect = 2, General = 3, large effect = 4, very large effect = 5 |
| Incentive regulation | Ere2 | How much is farmers' agricultural green production behaviors affected by government subsidy policies? No effect = 1, small effect = 2, general = 3, large effect = 4, very large effect = 5 |

**Table 1.** *Cont.*

| Variables | Number | Definition and Assignment |
|---|---|---|
| Guiding regulation | Ere3 | How much is farmers' agricultural green production behaviors affected by government technology extension policies? No effect = 1, small effect = 2, general = 3, large effect = 4, very large effect = 5 |
| Voluntary regulation | Ere4 | How much does the letter of commitment signed by the government and farmers affect farmers' agricultural green production behaviors? No effect = 1, small effect = 2, general = 3, large effect = 4, very large effect = 5 |
| Sex | Sex | Male = 1, female = 0 |
| Age | Age | [0, 30) = 1, [31, 40) = 2, [41, 50) = 3, [51, 60) = 4, [61, 70) = 5, [70, +∞) = 6 |
| Education level | Edu | Illiterate = 1, primary school = 2, junior middle school = 3, Senior school = 4, College or above = 5 |
| Arable area (mu) | Ara | [0, 3) = 1, [3, 5) = 2, [5, 10) = 3, [10, 15) = 4, [15, 20) = 5, [20, +∞) = 6 |
| Share of non-farm income | Nag | [0, 20%) = 1, [20%, 40%) = 2, [40%, 60%) = 3, [60%, 80%) = 4, [80%, 100%] = 5 |
| Household labor force | Lab | Number of labor force from families aged 16 and above: [1, 3] = 1, [4, 7] = 2, [8, +∞) = 3 |
| Effectiveness evaluation | Ete | Evaluation of the overall effect after the first implementation of agricultural green production behaviors? 1 = very bad, 2 = not so good, 3 = general, 4 = relatively good, 5 = very good |

1 ha ≈ 15 mu.

### 2.4. Economic Modeling

In the empirical analysis of the impacts of risk perception and environmental regulation on the farmers' sustainable behaviors of AGP, the farmers' willingness is a typical dichotomous discrete variable, and discrete selection models such as probit and logit are commonly used. Nevertheless, compared with the logit model's limitations in alternative forms and unobserved factors, The probit model can get rid of such troubles to a large extent and is more suitable for analyzing subject behaviors and predicting problems [28]. Therefore, this study assumes that the original data follow normal distribution and selects the binary probit model for further analysis [43,44]. The specific economic model is set as Equation (1):

$$P(Cin = 1|X) = \varnothing(\beta_0 + \beta_1 Rpe_i + \beta_2 Ere_i + \beta_3 Cva_i + \mu_i) \tag{1}$$

where *Cin* represents the sustainable behaviors of farmers in AGP, $Cin = 1$ represents willingness to continue the implementation and $Cin = 0$ represents unwillingness to continue the implementation. *Rpe* and *Ere* are the core explanatory variables, representing farmers' risk perception and government environmental regulation, respectively. *Cva* represents the control variable groups affecting the sustainable behaviors of farmers' AGP, including gender, age, education level, arable land area and other variables, and $\beta_0$ for the constants. $\beta_1$, $\beta_2$, $\beta_3$ are the coefficient estimation vectors of the regression model, $\varnothing$ is the probability function of normal distribution, and $\mu$ stands for the random error term. Furthermore, this study adopts the stepwise regression method to test the relationship between risk perception, environmental regulation and farmers' sustainable behaviors of AGP by controlling the selected control variables successively.

For the analysis of the moderating effect, this study adds the moderating variable and the product term of the moderating variable and core explanatory variable into Equation (1); that is, the product of the total index of environmental regulation and the total index of risk perception [39,45]. The equation of the specific model is shown in Equation (2):

$$Cin_i = \beta_1 Rpe_i + \beta_2 Ere_i + \beta_3 Cva_i + \beta_4 Rpe_i * Ere_i + \mu_i \tag{2}$$

For each dimension of the core explanatory variables, the same method is adopted as in Equation (2), namely, adding the interaction term multiplied by each dimension of risk perception and environmental regulation. The equations of the specific model are shown as Equations (3)–(6):

$$Cin_i = \alpha_j Rpe_1 + \beta_j Ere_i + \gamma_j Rpe_1 * Ere_i + \delta_j Cva_i + \mu_i \tag{3}$$

$$Cin_i = \alpha_j Rpe_2 + \beta_j Ere_i + \gamma_j Rpe_2 * Ere_i + \delta_j Cva_i + \mu_i \tag{4}$$

$$Cin_i = \alpha_j Rpe_3 + \beta_j Ere_i + \gamma_j Rpe_3 * Ere_i + \delta_j Cva_i + \mu_i \tag{5}$$

$$Cin_i = \alpha_j Rpe_4 + \beta_j Ere_i + \gamma_j Rpe_4 * Ere_i + \delta_j Cva_i + \mu_i \tag{6}$$

In Equations (3)–(6), $Rpe_1$, $Rpe_2$, $Rpe_3$, $Rpe_4$ represent economic risks, time risks, environmental risks and pressure risks, respectively, and $Rpe_1 * Ere_i$, $Rpe_2 * Ere_i$, $Rpe_3 * Ere_i$, $Rpe_4 * Ere_i$ represent the product of each dimension of environmental regulation and economic risks, time risks, environmental risks and pressure risks, respectively. $\alpha_j$, $\beta_j$, $\gamma_j$ and $\delta_i$ are the regression coefficients.

### 2.5. Sample Description

As shown in Table 2, among the 652 sample households investigated in this study, 478 households (73.313%) are male, and 174 households (26.687%) are female. In terms of age, 446 households (68.405%) were middle-aged and elderly households, with the average age of 51~60 years old being the most. In terms of education level, most of the households (546, accounting for 83.742% of the sample) were junior middle school or below, and a few households (106, accounting for 15.258% of the sample) were senior high school or above. In terms of arable land area, 354 households (69.632%) own an area of less than 10 mu, with an average of 7.342 mu. In terms of the non-farm income, the majority of farmers (589, accounting for 90.338% of the sample) have 40% or more of non-farm income. In terms of household labor force, there are 441 households (67.638%) with a family labor force of 1–3 persons, and the average household labor force of the whole sample households is 2.221 persons. In terms of effectiveness evaluation, farmers whose evaluation was very bad, not so good, average, relatively good and very good accounted for 1.074%, 6.442%, 17.484%, 41.718% and 33.282%, respectively, with an average value of 3.642. Among them, 489 households (75%) rated the overall effect of the initial implementing AGP behaviors as good or above.

**Table 2.** Description of the basic characteristics of the sample farmers (*n* = 652).

| Variable Name | Variable Number | Frequency | Percentage (%) | Mean | Standard Deviation |
|---|---|---|---|---|---|
| Sex | Sex | | | 0.733 | 0.443 |
| Male | | 478 | 73.313 | | |
| Female | | 174 | 26.687 | | |
| Age | Age | | | 45.801 | 10.566 |
| [0, 30) | | 44 | 6.748 | | |
| [31, 40) | | 162 | 24.847 | | |
| [41, 50) | | 138 | 21.166 | | |
| [51, 60) | | 191 | 29.294 | | |
| [61, 70) | | 96 | 14.724 | | |
| [70, +∞) | | 21 | 3.221 | | |
| Education | Edu | | | 2.271 | 1.217 |
| Illiteracy | | 194 | 29.754 | | |
| Primary school | | 253 | 38.804 | | |
| Junior middle school | | 99 | 15.184 | | |
| Senior school | | 59 | 9.049 | | |
| College or above | | 47 | 7.209 | | |

**Table 2.** *Cont.*

| Variable Name | Variable Number | Frequency | Percentage (%) | Mean | Standard Deviation |
|---|---|---|---|---|---|
| Arable area (mu) | Ara | | | 7.342 | 2.905 |
| [0, 3) | | 81 | 12.423 | | |
| [3, 5) | | 138 | 21.166 | | |
| [5, 10) | | 235 | 36.043 | | |
| [10, 15) | | 156 | 23.926 | | |
| [15, 20) | | 23 | 3.528 | | |
| [20, +∞) | | 19 | 2.914 | | |
| Share of non-farm income (%) | Nag | | | 3.880 | 1.015 |
| [0, 20) | | 11 | 1.687 | | |
| [20, 40) | | 52 | 7.975 | | |
| [40, 60) | | 157 | 24.080 | | |
| [60, 80) | | 210 | 32.209 | | |
| [80, 100] | | 222 | 34.049 | | |
| Household Labor force | Lab | | | 2.221 | 0.367 |
| [1, 3] | | 441 | 67.638 | | |
| [4, 7] | | 198 | 30.368 | | |
| [8, +∞) | | 13 | 1.994 | | |
| Effectiveness evaluation | Ete | | | 3.642 | 0.929 |
| Very bad | | 7 | 1.074 | | |
| Not so good | | 42 | 6.442 | | |
| General | | 114 | 17.484 | | |
| Relatively good | | 272 | 41.718 | | |
| Very good | | 217 | 33.282 | | |

1 ha ≈ 15 mu; USD 1 ≈ CNY 6.355.

Table 3 shows the descriptive statistics of farmers' sustainable behaviors of AGP and its core explanatory variables, namely risk perception and environmental regulation variables. It can be seen that 62.730% of farmers are willing to continue to implement AGP. However, there are also 37.270% farmers not willing to continue the implementation, which cannot be ignored.

In terms of risk perception variable, the total mean value is 3.908, indicating that farmers have a high level of risk perception brought by the sustainable behaviors of AGP. Among them, the mean score of economic risks dimension, namely "do you worry that more money will be spent to implement the agricultural green production behaviors", was the highest (mean = 4.280), followed by the time risks dimension (mean = 4.112), namely "do you worry that it will cost more time and energy to implement the agricultural green production behaviors". However, the mean score of stress risks dimension, namely "do you worry that your neighbors will talk more about implementing the agricultural green production behaviors" (mean = 3.439), was the lowest. These indicate that in the specific dimension of risk perception variable that affect farmers' sustainable behaviors, farmers are more worried about money input and time consumption, rather than surrounding pressure. However, the average value of the dimension of environmental risks is 3.800, that is, "do you worry that more money will be spent to implement the agricultural green production behaviors", which means that farmers' perception of environmental risks the continuous implementation of AGP is at a medium level, and their environmental awareness has been improved in recent years.

In terms of environmental regulation variable, the total mean value is divided into 3.642, indicating that the impact of government environmental regulation on farmers' sustainable behaviors of AGP is remarkable on the whole. In the specific dimension, voluntary regulation has the highest mean score (mean = 4.262), followed by guiding regulation (mean = 3.986), indicating that signing the commitment letter has a relatively large impact on farmers' continuous implementation of AGP and their behaviors are affected by the government's technology promotion policies. Nonetheless, the mean

score of constrained regulations is relatively low (mean = 3.420), which means that the sustainable behaviors of farmers' AGP is not much affected by government regulatory policies. Additionally, the mean score of incentive regulation is the lowest (mean = 2.919), and most farmers (84.970%) think that the score of "how much is farmers' agricultural green production behaviors affected by government subsidy policies" is 3 or below, indicating that the impacts of government subsidy policies on farmers' sustainable behaviors are not remarkable.

**Table 3.** Descriptive statistics of the explained variables and core explanatory variables (*n* = 652).

| Variables | Number | Assignment | Frequency | Percentage (%) | Mean | Standard Deviation |
|---|---|---|---|---|---|---|
| Willingness to continue | Cin | – | – | – | 0.627 | 0.484 |
| | | 1 | 409 | 62.730 | | |
| | | 0 | 243 | 37.270 | | |
| Risk perception | Rpe | – | – | – | 3.908 | 0.727 |
| Economic risks | Rpe1 | | | | 4.280 | 1.334 |
| | | 1 | 26 | 3.988 | | |
| | | 2 | 100 | 15.337 | | |
| | | 3 | 106 | 16.258 | | |
| | | 4 | 227 | 34.816 | | |
| | | 5 | 193 | 29.601 | | |
| Time risks | Rpe2 | | | | 4.112 | 1.063 |
| | | 1 | 13 | 1.994 | | |
| | | 2 | 57 | 8.742 | | |
| | | 3 | 101 | 15.491 | | |
| | | 4 | 261 | 40.031 | | |
| | | 5 | 220 | 33.742 | | |
| Environmental risks | Rpe3 | | | | 3.800 | 1.220 |
| | | 1 | 58 | 8.896 | | |
| | | 2 | 122 | 18.711 | | |
| | | 3 | 103 | 15.798 | | |
| | | 4 | 203 | 31.135 | | |
| | | 5 | 166 | 25.46 | | |
| Stress risks | Rpe4 | | | | 3.439 | 1.435 |
| | | 1 | 29 | 4.448 | | |
| | | 2 | 140 | 21.472 | | |
| | | 3 | 98 | 15.031 | | |
| | | 4 | 190 | 29.141 | | |
| | | 5 | 195 | 29.908 | | |
| Environmental regulation | Ere | – | – | – | 3.642 | 0.929 |
| | | 1 | 9 | 1.381 | | |
| | | 2 | 129 | 19.785 | | |
| | | 3 | 152 | 23.313 | | |
| | | 4 | 303 | 46.472 | | |
| | | 5 | 59 | 9.049 | | |
| Constrained regulation | Ere1 | | | | 3.420 | 0.951 |
| | | 1 | 9 | 1.381 | | |
| | | 2 | 129 | 19.785 | | |
| | | 3 | 152 | 23.313 | | |
| | | 4 | 303 | 46.472 | | |
| | | 5 | 59 | 9.049 | | |
| Incentive regulation | Ere2 | | | | 2.919 | 0.701 |
| | | 1 | 21 | 3.221 | | |
| | | 2 | 117 | 17.945 | | |
| | | 3 | 416 | 63.804 | | |
| | | 4 | 90 | 13.803 | | |
| | | 5 | 8 | 1.227 | | |

**Table 3.** *Cont.*

| Variables | Number | Assignment | Frequency | Percentage (%) | Mean | Standard Deviation |
|---|---|---|---|---|---|---|
| Guiding regulation | Ere3 | | | | 3.968 | 0.817 |
| | | 1 | 10 | 1.534 | | |
| | | 2 | 34 | 5.215 | | |
| | | 3 | 56 | 8.589 | | |
| | | 4 | 418 | 64.110 | | |
| | | 5 | 134 | 20.552 | | |
| Voluntary regulation | Ere4 | | | | 4.262 | 1.111 |
| | | 1 | 9 | 1.381 | | |
| | | 2 | 75 | 11.503 | | |
| | | 3 | 66 | 10.123 | | |
| | | 4 | 92 | 14.110 | | |
| | | 5 | 410 | 62.883 | | |

1–5 represents the degree of farmers' agreement with the items of each variable.

## 3. Results

This study uses Stata15.0 statistical software to analyze the impacts of risk perception and environmental regulation on the sustainable behaviors of farmers' AGP. Before regression, it is necessary to test whether multicollinearity exists between variables to ensure the accuracy and stability of each model. The results show that the values of tolerance and variance inflation factor (VIF) of all variables are within the reasonable ranges (Tol > 0.1, VIF < 5). Therefore, the explanatory variables can be successively incorporated into the binary probit model to estimate the regression results.

### 3.1. Baseline Regression Results

Table 4 shows the baseline regression results. Firstly, the impacts of risk perception and environmental regulation on the sustainable behaviors of farmers' AGP are separately discussed, and the estimated results are shown in Model 1. Secondly, eight two-dimensional variables, including economic risks, time risks, environmental risks and pressure risks, as well as constrained regulation, incentive regulation, guiding regulation and voluntary regulation, were introduced to further analyze the impacts of different dimensions, and the estimated results are shown in model 2. Finally, from the values of Log-likelihood, chi-square and Pseudo-R2, it can be seen that the overall fitting effect of each model is good.

**Table 4.** Regression estimation results of explanatory variables (*n* = 652).

| Variables | Model 1 | | Model 2 | |
|---|---|---|---|---|
| | Regression Results | Marginal Effects | Regression Results | Marginal Effects |
| Risk perception | −0.380 *** | −0.130 *** | | |
| | (0.096) | (0.032) | | |
| Economic risks | | | −0.216 ** | −0.073 ** |
| | | | (0.112) | (0.037) |
| Time risks | | | −0.108 * | −0.036 * |
| | | | (0.045) | (0.015) |
| Environmental risks | | | 0.079 * | 0.027 * |
| | | | (0.046) | (0.015) |
| Stress risks | | | −0.026 | −0.012 |
| | | | (0.041) | (0.014) |
| Environmental regulation | 0.691 *** | 0.237 *** | | |
| | (0.118) | (0.037) | | |
| Constrained regulation | | | 0.261 ** | 0.088 ** |
| | | | (0.056) | (0.018) |

**Table 4.** *Cont.*

| Variables | Model 1 | | Model 2 | |
|---|---|---|---|---|
| | **Regression Results** | **Marginal Effects** | **Regression Results** | **Marginal Effects** |
| Incentive regulation | | | 0.088 | 0.030 |
| | | | (0.063) | (0.021) |
| Guiding regulation | | | 0.273 ** | 0.092 ** |
| | | | (0.187) | (0.063) |
| Voluntary regulation | | | 0.417 *** | 0.141 *** |
| | | | (0.208) | (0.070) |
| Sex | 0.408 *** | 0.140 *** | 0.420 *** | 0.142 *** |
| | (0.120) | (0.040) | (0.122) | (0.040) |
| Age | 0.056 | 0.019 | 0.078 | 0.026 |
| | (0.053) | (0.018) | (0.055) | (0.018) |
| Education level | −0.018 | −0.006 | −0.001 | −0.001 |
| | (0.055) | (0.019) | (0.056) | (0.019) |
| Arable area | 0.117 ** | 0.040 ** | 0.111 ** | 0.037 ** |
| | (0.048) | (0.016) | (0.049) | (0.016) |
| Share of non-farm income | 0.121 ** | 0.042 ** | 0.123 ** | 0.042 ** |
| | (0.057) | (0.019) | (0.057) | (0.019) |
| Household labor force | 0.0375 | 0.013 | 0.0401 | 0.014 |
| | (0.033) | (0.011) | (0.034) | (0.012) |
| Effectiveness evaluation | 0.155 ** | 0.053 ** | 0.267 ** | 0.090 ** |
| | (0.063) | (0.022) | (0.126) | (0.042) |
| Pseudo-R2 | 0.090 | — | 0.101 | — |
| Log-likelihood | −391.994 | — | −387.084 | — |
| Prob > chi2 | 0.000 | — | 0.000 | — |
| LR-test | 77.140 | — | 86.960 | — |
| Observations | 652 | — | 652 | — |

*, **, and *** was significant at the level of 1%, 5% and 10%, respectively.

### 3.1.1. Influence of Risk Perception

The estimation results of model 1 show that risk perception has a significantly negative effect at $p < 0.001$ (regression coefficient is −0.380), and the marginal effect was −0.130. This shows that the probability of farmers' willingness to continue to implement AGP will decrease by 13% when their risk perception increases by one unit. Specifically, the estimation results of Model 2 reveal that two-dimensional variables, namely economic risks and time risks, have significantly negative effects at $p < 0.01$ and $p < 0.05$, respectively, while environmental risks have significantly positive effects at $p < 0.05$. Among them, the economic risks have the greatest influence on the sustainable behaviors of farmers' AGP (regression coefficient is −0.216), followed by the time risks (regression coefficient is −0.108). When other conditions remain the same, the possibility of farmers' willingness to continue to implement AGP will be reduced by 7.3% and 3.6%, respectively, because they worry that implementing the AGP will cost more money, spend more time and energy. At the same time, the possibility of farmers' willingness to continue to implement the AGP will increase by 2.7% with each unit increase, because they believe that no longer implementing AGP will cause more serious damage to the eco-environment. Stress risks have no significantly negative impacts on farmers' sustainable behaviors of AGP, indicating that farmers are not worried about whether their sustainable behaviors will be discussed by their neighbors.

### 3.1.2. Influence of Environmental Regulation

Model 1 shows that environmental regulation has a significantly positive effect at $p < 0.001$ (regression coefficient is 0.691), and the marginal effect is 0.237. This shows that the probability of continuing to implement the AGP will increase by 23.7% when the influence degree of government environmental regulation increases by 1 unit. Specifically, the estimation results of Model 2 reveal that voluntary regulation has a significantly positive

effect at $p < 0.001$, and it has the greatest impact on the willingness of farmers to continue to implement the AGP (regression coefficient is 0.417), followed by guiding regulation and constrained regulation, and the number of regression coefficients are 0.273 and 0.261, respectively. When other conditions are controlled, the probability of farmers' willingness to continue to implement the AGP will increase by 14.1% when the influence degree of the commitment letter signed by the government and farmers increases by one unit. The probability of farmers' willingness to continue to implement the AGP will increase by 9.2% when the influence degree of government technology promotion policies increases by one unit, while the probability will increase by 8.8% when the influence degree of government supervision policies increases by one unit. However, the positive impacts of government subsidy policies are not significant. This may be related to the low level of subsidies and inadequate distribution, as well as the poor effects of environmental incentive measures adopted by the government.

### 3.1.3. Influence of Control Variables

In the two models shown in Table 4, the control variables of gender, arable area and share of non-farm income have significantly positive effects on the sustainable behaviors of farmers' AGP at $p < 0.001$, $p < 0.01$ and $p < 0.05$, respectively. Age and household labor force also had a positive effect, but not significant. Moreover, the negative effect of education level is not significant. The effectiveness evaluation has a significant positive impact at $p < 0.01$. Because when farmers perceive that the implementation of AGP behaviors is more beneficial than harmful, it will promote the internal motivation of farmers' willingness. This is basically consistent with the evaluation results of the overall effects of the first implementation of AGP behaviors.

### 3.2. Moderating Role of Environmental Regulation

In order to verify the moderating effect of environmental regulation on risk perception and sustainable behaviors of farmers' AGP, the regression models in Table 5 add the interaction items multiplied by risk perception (including its dimensions) and environmental regulation (including its dimensions) on the basis of the benchmark regression. Table 5 only shows the product items that pass the significance test after screening. The estimation results of model 1 show that the product term of environmental regulation and risk perception is significant at $p < 0.001$, and the regression coefficient is 0.253. This indicates that environmental regulation positively regulates farmers' risk perception, and then promotes farmers' willingness to continue the implementation of AGP behaviors.

The estimation results from Model 2 to Model 8 further reveal the functions and relationships among variables. Specifically, the product terms of constrained regulation and economic risks, time risks and environmental risks are significant at $p < 0.001$, $p < 0.01$ and $p < 0.001$, respectively, and the regression coefficients are 0.043, 0.047 and 0.055, respectively. The results show that constrained regulation has significantly positive moderating effect on farmers' economic risks, time risks and environmental risks and their sustainable behaviors of AGP. Moreover, restrictive regulations can, to a certain extent, enhance the perception of environmental risks that "farmers will damage the agricultural production environment if they do not continue to implement the agricultural green production behaviors" through regulatory punishment and other restrictive behavior measures in order to promote farmers' willingness to continue to implement the AGP.

The product terms of voluntary regulation and economic risks, time risks, environmental risks and pressure risks are significant at $p < 0.001$ level, and the regression coefficients are 0.112, 0.118, 0.073 and 0.062, respectively. This shows that voluntary regulation has a significantly positive moderating effect on the economic risks, time risks, environmental risks and pressure risks and their sustainable behaviors of AGP. Moreover, voluntary regulation can better reduce the time risks, and thus encourage farmers to spend more time and energy on the continuous implementation of AGP behaviors. It can be seen from the estimation results of the regression model above that both voluntary regulation and

constrained regulation play a certain regulatory role, yet the effect of voluntary regulation is stronger than that of constrained regulation on the whole. Because the environmental regulation policies signed by the government and farmers through the commitment letter can better stimulate farmers' willingness to continue to implement the AGP behaviors.

**Table 5.** Regression results of moderating effects from environmental regulation and its dimensions (*n* = 652).

| Variables | Model 1 | Model 2 | Model 3 | Model 4 | Model 5 | Model 6 | Model 7 | Model 8 |
|---|---|---|---|---|---|---|---|---|
| Risk perception | −1.276 *** | | | | | | | |
| | (0.204) | | | | | | | |
| Risk perception* Environmental regulation | 0.253 *** | | | | | | | |
| | (0.049) | | | | | | | |
| Economic risks | | −0.254 *** | −0.588 *** | | | | | |
| | | (0.089) | (0.103) | | | | | |
| Economic risks* Constrained regulation | | 0.043 ** | | | | | | |
| | | (0.016) | | | | | | |
| Economic risks* Voluntary regulation | | | 0.112 *** | | | | | |
| | | | (0.021) | | | | | |
| Time risks | | | | −0.382 *** | −0.757 *** | | | |
| | | | | (0.137) | (0.155) | | | |
| Time risks* Constrained regulation | | | | 0.047 *** | | | | |
| | | | | (0.022) | | | | |
| Time risks* Voluntary regulation | | | | | 0.118 *** | | | |
| | | | | | (0.023) | | | |
| Economic risks | | | | | | 0.231 *** | 0.379 *** | |
| | | | | | | (0.067) | (0.071) | |
| Economic risks* Constrained regulation | | | | | | 0.055 *** | | |
| | | | | | | (0.025) | | |
| Economic risks* Voluntary regulation | | | | | | | 0.073 *** | |
| | | | | | | | (0.014) | |
| Stress risks | | | | | | | | −0.372 *** |
| | | | | | | | | (0.085) |
| Stress risks* Voluntary regulation | | | | | | | | 0.062 *** |
| | | | | | | | | (0.018) |
| Sex | 0.395 *** | 0.372 *** | 0.390 *** | 0.372 *** | 0.366 *** | 0.355 *** | 0.351 *** | 0.380 *** |
| | (0.119) | (0.117) | (0.119) | (0.117) | (0.118) | (0.117) | (0.119) | (0.118) |
| Age | 0.066 | 0.066 | 0.054 | 0.076 | 0.07 | 0.0684 | 0.0704 | 0.062 |
| | (0.053) | (0.052) | (0.052) | (0.052) | (0.052) | (0.052) | (0.052) | −0.052 |
| Education level | −0.009 | −0.007 | −0.005 | −0.005 | −0.001 | −0.0103 | −0.004 | −0.016 |
| | (0.055) | (0.053) | (0.054) | (0.054) | (0.054) | (0.054) | (0.054) | −0.054 |
| Arable area | 0.111 ** | 0.0904 * | 0.099 ** | 0.096 ** | 0.104 ** | 0.099 ** | 0.099 ** | 0.097 ** |
| | (0.048) | (0.047) | (0.048) | (0.047) | (0.047) | (0.047) | (0.048) | −0.047 |
| Share of non-farm income | 0.133 ** | 0.147 *** | 0.128 ** | 0.151 *** | 0.130 ** | 0.141 ** | 0.121 ** | 0.138 ** |
| | (0.056) | (0.055) | (0.055) | (0.055) | (0.056) | (0.055) | (0.055) | −0.056 |
| Household labor force | 0.0367 | 0.0431 | 0.0441 | 0.0313 | 0.0311 | 0.028 | 0.029 | 0.041 |
| | (0.033) | (0.033) | (0.033) | (0.032) | (0.033) | (0.032) | (0.033) | −0.033 |
| Effectiveness evaluation | −0.154 ** | −0.0948 | 0.080 | 0.280 ** | 0.309 ** | 0.0758 | 0.054 | 0.072 |
| | (0.064) | (0.059) | (0.060) | (0.125) | (0.124) | (0.059) | (0.060) | −0.059 |
| Pseudo-R2 | 0.081 | 0.042 | 0.072 | 0.042 | 0.066 | 0.046 | 0.068 | 0.057 |
| Log-likelihood | −395.925 | −412.462 | −399.597 | −412.709 | −402.251 | −410.969 | −401.21 | −406.249 |
| LR-test | 69.28 | 36.21 | 61.94 | 35.71 | 56.63 | 39.19 | 58.71 | 48.63 |
| Observations | 652 | 652 | 652 | 652 | 652 | 652 | 652 | 652 |

Due to space limitations, this article only reports the important parts; *, ** and *** were significant at the level of 1%, 5% and 10%, respectively.

### 3.3. Robustness Test

In order to test the robustness of the estimated results, this study, referring to the research methods of Gong et al. (2016) [23] and Li et al. (2021) [45], tests the robustness of the regression results in Sections 3.1 and 3.2 by deleting the sample data with too old and too young ages. To be specific, the elderly samples >60 years old and adolescents <18 years old are deleted from 652 samples, and Logit model is used for regression to test the robustness of the above results. According to the comparison of Tables 4–6, it can be seen that the significance of farmers' risk perception, government environmental regulation,

their two-dimensional variables and product terms are basically the same, which indicates that the robustness of the model is verified.

**Table 6.** Robustness test results (*n* = 652).

| Variables | Model 1 | Model 2 | Model 3 | Model 4 | Model 5 | Model 6 | Model 7 | Model 8 | Model 9 |
|---|---|---|---|---|---|---|---|---|---|
| Risk perception | −0.003 ** (0.024) | | −0.064 ** (0.029) | | | | | | |
| Economic risks | | −0.031 *** (0.018) | | −0.232 *** (0.043) | | | | | |
| Time risks | | −0.164 *** (0.050) | | | −0.283 *** (0.085) | −0.383 *** (0.067) | | | |
| Environmental risks | | 0.023 (0.018) | | | | | 0.064 ** (0.029) | 0.152 *** (0.030) | |
| Stress risks | | −0.027 * (0.014) | | | | | | | −0.147 *** (0.032) |
| Environmental regulation | 0.001 ** (0.021) | | | | | | | | |
| Constrained regulation | | 0.001 (0.026) | | | | | | | |
| Incentive regulation | | 0.094 (0.079) | | | | | | | |
| Guiding regulation | | 0.072 (0.071) | | | | | | | |
| Voluntary regulation | | 0.118 *** (0.023) | | | | | | | |
| Risk perception* Environmental regulation | | | 0.014 ** (0.007) | | | | | | |
| Economic risks* Voluntary regulation | | | | 0.046 *** (0.009) | | | | | |
| Time risks* Constrained regulation | | | | | 0.014 * (0.007) | | | | |
| Time risks* Voluntary regulation | | | | | | 0.047 *** (0.010) | | | |
| Economic risks* Constrained regulation | | | | | | | 0.025 ** (0.015) | | |
| Economic risks* Voluntary regulation | | | | | | | | 0.033 *** (0.006) | |
| Stress risks* Voluntary regulation | | | | | | | | | 0.031 *** (0.007) |
| Control variables | Controlled | Controlled | Controlled | Controlled | Controlled | Controlled | Controlled | Controlled | Controlled |
| Observations | 510 | 510 | 510 | 510 | 510 | 510 | 510 | 510 | 510 |
| R-squared | 0.112 | 0.125 | 0.052 | 0.097 | 0.068 | 0.104 | 0.052 | 0.096 | 0.081 |

*, **, and *** was significant at the level of 1%, 5% and 10%, respectively.

## 4. Discussion

This study investigates the sustainable behaviors of farmers' AGP in three super grain producing counties in Henan Province, China. Results show that the majority of farmers in the investigated areas are willing to continue to implement the AGP behaviors. However, there are also nearly 40% of farmers are not willing to, which cannot be ignored. In order to identify the key factors influencing farmers' sustainable behaviors, this study recombined and optimized the dimensions of farmers' risk perception and government environmental regulation and unified the two core explanatory variables to investigate the impacts of risk perception and environmental regulation and their dimensions on the sustainable behaviors of farmers' AGP. This breaks through the limitation of previous studies that only one or two indicators are selected separately for discussion. This study further discusses the moderating effect of environmental regulation and its dimensions on risk perception and farmers' sustainable behaviors. Moreover, the results show that the model has passed the robustness test, indicating that the data analysis results are relatively reliable and stable.

This study finds that the higher the farmers' risk perception, the less willing they are to continuously implement AGP behaviors, which has been confirmed in some relevant literatures [28,29]. Because farmers are rational economic men, they generally have risk

aversion when facing behavioral decisions [46]. Li et al. (2020a) [8] also pointed out that, to a large extent, farmers' risk perception would inhibit the conversion of farmers' willingness to their actual green disposal behaviors concerning the disposal of pesticide packaging waste, or even show a state of "inadequate". Moreover, according to the estimation results of Model 2 in Table 4, among all dimensions of risk perception, economic risks have the greatest impacts on farmers' willingness, and the direction is negative. The reason is that the farmers worry that the continuous implementation of AGP requires additional investment of more money, which will bring economic burden to the farmers themselves and their families. Meanwhile, AGP is difficult to obtain considerable benefits in a short period of time [16], so that farmers think it is better to work outside for a long time to earn money than to continue to implement the AGP. This study also finds that environmental risks have a significantly positive impact on farmers' willingness, indicating that farmers' environmental awareness level in agricultural production activities has been improved in recent years. Farmers actively recognize that implementing AGP behaviors can reduce environmental pollution and benefit their health [5], which may be one of the important reasons to actively promote farmers' willingness to continue the implementation of AGP behaviors.

In this study, environmental regulation variable has a significant positive impact on the sustainable behaviors, which is different from the impact direction of risk perception variable. Specifically, the greater the impacts of restrictive regulations, guiding regulations and voluntary regulations adopted by the government, the more likely farmers will continue to implement AGP behaviors. This is basically consistent with the conclusions of Teklewold et al. (2013) [41], Ren et al. (2018) [47] and Dessart et al. (2019) [48]. Because the stronger the government regulates farmers' pesticide abuse, litter and other behaviors through supervision and punishment, the more it can promote farmers to move closer to the restrictive regulation goal [45]. In addition, effective publicity and training, promotion and demonstration can make farmers gradually realize the economic value and environmental benefits of AGP behaviors, so as to guide farmers to adjust and standardize their production behaviors according to the requirements of green regulation [49,50]. These conclusions, based on the above reasons, explain the improvement effect of restrictive and guiding regulations on farmers' willingness to continue to implement AGP behaviors from different perspectives.

Nevertheless, this study finds that among the dimensions of environmental regulation, voluntary regulation has the greatest impacts on farmers' willingness to continue the implementation. Because the environmental agreement signed by the government and farmers can drive farmers to take the initiative. The field research data analysis results in Table 3 show that the average value of voluntary regulations is the highest (mean = 4.262), which strongly supports the above views. Whereas the influence of incentive regulation in this study is not significant, which is different from the conclusions drawn by some researchers. For example, Chen et al. (2017) [51] pointed out that market-oriented incentive means such as material reward, economic subsidy and tax exemption can stimulate farmers to choose good behaviors of reduction, harmlessness and resource recovery. However, Puntsagdorj et al. (2021) [52] confirmed that economic incentive regulations such as subsidy policies have no significant impact on wheat growers' environmental protection behaviors of adopting sustainable agricultural practices. This may be related to the low level of subsidies and inadequate distribution, which makes it difficult to induce farmers' motivation for environmental protection [32]. Meanwhile, it also shows that the effect of incentive regulation policies or measures adopted by the government on farmers' willingness to continue to implement AGP behaviors has not reached the expected value.

In addition, as one of the typical contents of social regulation, the government can regulate economic behaviors by formulating corresponding environmental regulation policies and measures [53], so as to achieve the goal of coordination between environmental protection and economic development [54]. Therefore, for agricultural production, this study also focuses on the impacts of the product term of risk perception and environmen-

tal regulation. It is found that there is a significant interaction between environmental regulation and risk perception, that is, environmental regulation positively regulates risk perception and promotes farmers' willingness to continue the implementation. This conclusion is supported by a study on the impact of farmers' psychological perception on their green fertilization behaviors [55], because the psychological perception of farmers will have a differentiated impact on their decision-making behaviors due to the difference of environmental regulations. In terms of multiplying two variables in the two dimensions, this study finds that constrained regulations have the strongest moderating effect on farmers' environmental risks and sustainable behaviors of the AGP. The government is more able to enhance farmers' environmental risks through regulatory punishment and other restrictive behavioral measures [56], thus promoting farmers' willingness to continue the implementation. However, voluntary regulation has the strongest moderating effect on farmers' time risks and sustainable behaviors of the AGP. Because the government is more able to adopt contractual regulation measures by signing commitment letters with farmers to urge behavior subjects to compromise with each other and reach cooperative intentions [57,58], so as to drive other surrounding farmers to voluntarily participate in green production. Si et al. (2020) [39] emphasizes the above views and believes that voluntary regulation can promote farmers to resist improper disposal behaviors and take the initiative to adopt resource disposal technologies by improving their risk cognitive level.

This study can provide some important policy implications. First, promoting the AGP technology innovation can accelerate the realization of green input factors cost saving and efficiency. Meanwhile, adhering to the principle of endogenous drive, constantly reducing the perception of economic risks can encourage farmers to spend money, time and energy to continue to implement AGP behaviors. Second, the governance contract model in which the government signs the commitment letters with farmers should be formulated and improved. New business entities such as large farmers should be cultivated in terms of policy support, production guidance and input guarantee, so as to farmers including ordinary smallholders will be encouraged to be willing to engage in the sustainable AGP behaviors. Third, the sustainable behaviors can be induced and encouraged by the constraint system through strengthening the construction of corresponding responsibility supervision mechanism, such as imposing a specified amount of fine, or ordering farmers to abandon non-green production behaviors. Meanwhile, voluntary and constrained regulation should also be coordinated to play an enhanced regulatory role, and then the effects of the two-pronged regulation policies can be improved to maximize the promotion of farmers' AGP. Finally, other developing countries like China are also faced with excessive use of chemical inputs and improper disposal of agricultural waste in the agricultural production. The conclusions in this study can provide important reference value for other developing countries to adopt relevant policies to prevent and control environmental pollution in agricultural production and promote the farmers' sustainable AGP behaviors.

## 5. Conclusions, Limitations, and Future Research

The key to solve the prominent environmental pollution problem and achieve high-quality agricultural development is to encourage farmers to carry out sustainable AGP behaviors. In order to explore this topic, this study analyzed the impacts of risk perception and environmental regulation on farmers' sustainable behaviors of AGP based on the survey data. Results showed that both risk perception and environmental regulation have significant impacts on farmers' willingness to continue implementation. Moreover, economic risks have the greatest negative impacts in terms of risk perception followed by time risks. As for environmental regulation, voluntary regulation has the strongest positive effect, followed by guiding regulation and constrained regulation, yet incentive regulation has no significant positive effect. Besides, environmental regulation has a significantly positive moderating effect between the impacts of risk perception and farmers' willingness to implement AGP behaviors. On the whole, constrained regulation and

voluntary regulation have enhanced moderating effects, yet voluntary regulation has a more remarkable effect than constrained regulation.

This study has made in-depth exploration and good progress, which provides new insights to promote farmers' willingness to continue to implement AGP behaviors. Nevertheless, some limitations of this study should be acknowledged. Firstly, this study focuses on the direct impacts of risk perception and environmental regulation and their product terms on the sustainable behaviors of farmers' AGP. Other possible mediating factors are not involved, such as technology acquisition ability [59] and policy satisfaction evaluation [5]. Secondly, this study does not distinguish the continuous implementation willingness of farmers of different groups, considering that they can be driven by different factors [45]. Thirdly, only the grain-growing farmers in Henan Province were involved. Whether the findings can be generalized to other parts of China remains an open question. Fourthly, this paper uses micro-survey data obtained by visiting households at a time point, without considering panel data with time trend.

Future research can be expanded in the following aspects. Firstly, more mediating variables like technology acquisition ability can be appropriately selected. Secondly, the heterogeneity of farmers' willingness to implement AGP behaviors should be involved in different groups. Thirdly, the investigation area should be expanded. For instance, more grain farmers in other big grain-producing provinces can be involved, so as to make the research conclusion more universal and popularization significance. Finally, future research can obtain intertemporal survey data at the national level in a longer period of time to better analyze the sustainable behaviors.

**Author Contributions:** Conceptualization, M.L., K.C. and Y.L.; Data curation, M.L. and Y.L.; Funding acquisition, K.C.; Formal analysis, M.L. and Y.H.; Investigation, M.L. and Y.L.; Methodology, M.L. and L.W.; Supervision, M.L. and K.C.; Writing—original draft preparation, M.L., Y.L. and K.C.; Writing—review and editing, M.L., Y.L. and K.C. All authors have read and agreed to the published version of the manuscript.

**Funding:** This research was supported by Major Projects of The National Social Science Fund of China, grant number [20ZDA087] and Innovation and Entrepreneurship Training program for College students of Beijing Forestry University, grant number [X202110022115].

**Institutional Review Board Statement:** Not applicable.

**Informed Consent Statement:** Not applicable.

**Data Availability Statement:** The data presented in this study are available on request from the first author.

**Acknowledgments:** We are indebted to the anonymous reviewers and editors. We also give our special thanks to Xu Zhang who gave us precious advice.

**Conflicts of Interest:** The authors declare no conflict of interest.

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
