# Peer review of "Impacts of Risk Perception and Environmental Regulation on Farmers’ Sustainable Behaviors of Agricultural Green Production in China"

_agriculture, doi:10.3390/agriculture12060831_

Round 1

Reviewer 1 Report

Thank you for presenting a good and beneficial research. However, there are several advice to improve this paper as follows:

Please explain more detail your research gap including theoretical gap and empirical gap. It will be better if you can state clearly your research question.

Several current research will be beneficial to improve your references:

·       Strategy for Sustainability of Social Enterprise in Indonesia: A Structural Equation Modeling Approach. Sustainability, (2022)14, 1383.

·       Roles of regulation and lifestyle on Indonesian coffee consumption behavior across generations. International Journal of Sustainable Development and Planning, (2021) 16(6), pp. 1153–1162.

·       Business survival of small and medium-sized restaurants through a crisis: The role of government support and innovation. Sustainability, (2021)13(19), 10535.

Please clarify sampling method and how did you develop your variable.

In conclusion section, you don’t need to rewrite something that has been presented in result and discussion section. Policy implication is ok, however, you need to discuss the future research recommendation clearly.

Reviewer 2 Report

1.    1  Do you worry that more money will be spent to implement agricultural green production behaviors? Strongly disagree =1, strongly disagree =2, generally =3, somewhat agree =4, strongly agree =5

The same expression is used for 1 and 2 in Table 1 on page 6. The expression “strongly disagree =2” should be corrected as “disagree =2”.

2. In 5-point Likert scale questions, the expression in the middle should be neutral. Therefore, neutral or no idea should be used instead of 3=generally.

3. The study provides important data to measure the behavior of farmers engaged in agricultural green production and to guide environmental policies.

Author Response

Pls see the attachment.

Reviewer 3 Report

Some of the comments and suggestions to improve the paper:

(1) Since the study is related to China and micro-survey data set is collected from farmers in China, the title may be slightly modified to tell that it is for China only by adding “….in China” or “….of China” or “A Study of China’ at the end of the title.

(2) A stronger economic justification should be given for the selection of Henan Province as the study area. Please give some information on what is happening in other parts of China on AGP.

(3) Please give the theoretical justification behind the study with respect to AGP, preferably as a small section, before section 2 or inside first section itself.

(4) Items in section 3 and 4 are interesting. But please try to improve the overall result analysis and discussion in sections on results and discussion. Results and discussions are giving emphasis more on “as it is” or “what happens” based on data figures. Please try to explain in terms of “why it happens” also.

(5) Some of the sentences (like the first sentence in abstract) are very long. So, a long sentence may be split into two or three short sentences wherever possible without losing the appropriate meanings.

(6) Every study suffers from some limitations. But the present study has not given any limitation.

---x---

Author Response

Pls see the attachment.
